# Incidence of Surgical Site Infection and Use of Antibiotics among Patients Who Underwent Caesarean Section and Herniorrhaphy at a Regional Referral Hospital, Sierra Leone

**DOI:** 10.3390/ijerph19074048

**Published:** 2022-03-29

**Authors:** Ronald Carshon-Marsh, James Sylvester Squire, Kadijatu Nabbie Kamara, Aelita Sargsyan, Alexandre Delamou, Bienvenu Salim Camara, Marcel Manzi, Jamie Ann Guth, Mohamed Ahmed Khogali, Anthony Reid, Sartie Kenneh

**Affiliations:** 1District Health Management Team, Ministry of Health and Sanitation (MOHS), Bo District, Bo City 00232, Sierra Leone; 2National Disease Surveillance Programme, Directorate of Health Security and Emergencies, MOHS, Cockerill, Wilkinson Road, Freetown 00232, Sierra Leone; jmssquire@yahoo.com (J.S.S.); kamarakadijatunabie@gmail.com (K.N.K.); 3TB Research and Prevention Centre, Yerevan 0014, Armenia; sargsyan.aelita@gmail.com; 4Department of Public Health, Gamal University of Conakry, Conakry BP 1147, Guinea; adelamou@gmail.com; 5Centre National de Formation et de Recherche en Santé Rurale de Maferinyah, Maferinyah National Center for Training and Research in Rural Health, Forécariah BP 2649, Guinea; bienvenusalimcamara@gmail.com; 6Independent Researcher, 5000 Namur, Belgium; m.manzi449@gmail.com; 7Global Health Connections, Center Barnstead, Barnstead, NH 03225, USA; guth.jamie@gmail.com; 8Special Programme for Research and Training in Tropical Diseases (TDR), World Health Organisation, Avenue Appia 20, 1211 Geneva, Switzerland; khogalim@who.int; 9Operational Research Unit Luxembourg, Medecins Sans Frontieres/Doctors without Borders, 68 Rue Gasperich, L-1617 Luxembourg, Luxembourg; tony.reid@brussels.msf.org; 10Office of the Chief Medical Officer, Ministry of Health and Sanitation, 4th Floor, Youyi Building, Brookfields, Freetown 00232, Sierra Leone; sartiekenneh@gmail.com

**Keywords:** surgical site infection, surgical antibiotic prophylaxis, antibiotic use, caesarean section, herniorrhaphy, SORT IT, AMR, Sierra Leone

## Abstract

Surgical site infections (SSIs) are common postoperative complications. Surgical antibiotic prophylaxis (SAP) can prevent the occurrence of SSIs if administered appropriately. We carried out a retrospective cohort study to determine the incidence of SSIs and assess whether SAP were administered according to WHO guidelines for Caesarean section (CS) and herniorrhaphy patients in Bo regional government hospital from November 2019 to October 2020. The analysis included 681 patients (599 CSs and 82 herniorrhaphies). Overall, the SSI rate was 6.7% among all patients, and 7.5% and 1.2% among CS patients and herniorrhaphy patients, respectively. SAP was administered preoperatively in 85% of CS and 70% of herniorrhaphy patients. Postoperative antibiotics were prescribed to 85% of CS and 100% of herniorrhaphy patients. Ampicillin, metronidazole, and amoxicillin were the most commonly used antibiotics. The relatively low rate of SSIs observed in this study is probably due to improved infection prevention and control (IPC) measures following the Ebola outbreak and the current COVID-19 pandemic. A good compliance rate with WHO guidelines for preoperative SAP was observed. However, there was a high use of postoperative antibiotics, which is not in line with WHO guidelines. Recommendations were made to ensure the appropriate administration of SAP and reduce unnecessary use of antibiotics.

## 1. Introduction

A surgical site infection (SSI) is one that occurs after surgery at or near the surgical incision. SSIs can occur up to 30 days after surgery, or even up to a year in patients who have received implants [1]. The rate of SSI in low and middle-income countries ranges from 1% to 4%. These relatively low rates occurred over the past decade due to improved water, sanitation, and hygiene infrastructure, and relatively clean surgical operations [2].

Many of the most serious and difficult-to-treat antibiotic resistant infections occur in healthcare facilities. This is because of the intensive use of antibiotics and because severely ill patients with serious infections are admitted to health facilities. SSIs are the most common healthcare-associated infection, accounting for about 15% of all hospital-acquired infections [3,4,5]. Inadequate measures to prevent and control infection may contribute to the spread of microorganisms resistant to antimicrobial medicines [6]. One of the main strategies of the global action plan (GAP) to tackle antimicrobial resistance (AMR) is to reduce the incidence of all types of infection, including SSIs [6]. A key pillar in SSI prevention is the administration of surgical antibiotic prophylaxis (SAP). It is defined as administering an effective antimicrobial agent before exposure to contamination during surgery [7].

In Sub-Saharan Africa, Caesarean sections (CS) account for up to 80% of the surgical workload, with most (77%) being emergency surgeries that are more prone to developing SSIs [8]. In contrast, surgery performed under elective conditions has a reduced risk of SSI. According to Bayar et al. (2015), complications such as SSI were higher among patients in the emergency group (27%) than in the elective group (11%) [9]. A multi-country study by Medecins Sans Frontieres in 2015 among women who underwent CS in Burundi, the Democratic Republic of Congo (DRC), and Sierra Leone reported an SSI rate of 7.3% [8]. For CS, the World Health Organization (WHO) strongly recommends SAP be administered within 120 min prior to skin incision as a single dose of first-generation cephalosporin (cefazolin) or penicillin; ongoing use of antibiotics post surgery is not recommended. WHO also recommends a single dose of intravenous cefazolin just before the skin incision, with redosing required if the procedure takes longer than anticipated. However, clean procedures such as elective herniorrhaphy may not require SAP [7]. Reasons provided for poor compliance include the persistence of old practices, non-compliance with proper aseptic techniques, overcrowded hospitals, and poor knowledge of current correct antibiotic use [10].

WHO also recommends monitoring the rate of SSIs through surveillance in health facilities as a proxy indicator of maintaining standards of infection prevention and control (IPC) [9]. In Sierra Leone, there is no national guideline for the use of SAP. Therefore, the different aspects of administering these medications in terms of the choice of antibiotic, timing of administration, dose, and redose have been left entirely to the judgement of the operating surgeons. So far, there has been no formal assessment of SAP administration practice in Sierra Leone. Such information will inform antimicrobial stewardship programmes in hospitals. In addition, the only reported rate of SSIs came from one of the few documented studies published, in January 2020, from a tertiary referral hospital in Sierra Leone, where 1 in 10 (10.9%) women who underwent CS had SSI and 5.3% of these women eventually died [11]. Otherwise, there is virtually no current published literature on SSIs in Sierra Leone. There is a need for new SSI surveillance study so that we can follow the progress in terms of adherence to IPC practices.

Bo regional hospital has been documenting SSIs since 2019 in both the maternity operating theatre where CSs take place and in the main surgical theatre where elective cases are managed. CSs and herniorrhaphies make up the majority of cases in both theatres and antibiotics are prescribed freely in those situations. Thus, it was possible to document up-to-date SSI incidence in both high risk and elective cases, and to describe the current use of perioperative antibiotics in both contexts.

The aim of this study was to document the incidence of SSIs and the use of SAP in CS and herniorrhaphy patients in the Bo regional government hospital of Sierra Leone from November 2019 to October 2020. Specific objectives were to describe: (i) the incidence of SSIs in CS and herniorrhaphy patients; (ii) the type and proportion of antibiotics used; and (iii) the timing of antibiotics used for CSs and herniorrhaphies.

## 2. Materials and Methods

### 2.1. Study Design

This was a retrospective cohort study using routinely collected hospital data.

### 2.2. General Setting

Sierra Leone is in West Africa with a projected population of 7,100,000 according to the 2015 National Population and Housing Census [12]. It is bordered by Guinea in the north and east, the Atlantic Ocean in the west, and Liberia in the south. It has 16 districts divided into 190 chiefdoms.

The main challenges for the healthcare system in Sierra Leone are chronic underfunding, a heavy disease burden, and vastly insufficient numbers of skilled healthcare workers [13,14]. The Gross National Income (GNI) per capita (purchasing power parity in current dollars) was $490 for the year 2020 [15]. The total number of health facilities is about 1280, among which 24 are government hospitals, 27 are private hospitals, and 45 are private clinics [14].

### 2.3. Specific Setting

Bo District is in the southern province in Sierra Leone. It shares a boundary in the north with Tonkolili District, in the north-east with Kenema District, in the south-east with Pujehun District, in the south with Bonthe District, and in the west with Moyamba. The projected 2020 population, based on the 2015 census with a growth rate of 3.2% was 644,800. There are 142 peripheral health units (PHUs), 7 private/faith-based hospitals and 1 government tertiary referral hospital [14].

### 2.4. Study Site

The study was conducted at the Bo government referral hospital. The hospital has 308 beds and performs an average of 30 CSs per month. It has two major theatres, one in the maternity unit and the other in the main surgery complex which is called the main surgical theatre. SSI surveillance started in November 2019 at the maternity unit. The National IPC unit (NIPCU), supported by WHO, was responsible for establishing this SSI surveillance system. Various IPC measures have been implemented and strengthened at the Bo government hospital based on the NIPCU guidelines after the Ebola epidemic in 2014–2016 and reinforced during the COVID-19 pandemic. However, there are intermittent issues with the IPC supply chain and there is considerable inertia regarding attitudes toward IPC among the staff.

### 2.5. Study Population and Period

The surveillance study included all patients who underwent CS or herniorrhaphy from November 2019 to October 2020. Post-discharge surveillance was carried out for 30 days. After discharge, telephone calls were made by the unit staff to patients at least twice, after the end of the first-to-second week since the patients normally come to the hospital for dressing of their surgical wounds and removal of stitches. For those patients referred from distant communities, the staff in charge of the PHU performed the follow-up on the patient.

### 2.6. Use of Antibiotics

Prescribing SAP requires the correct indication, antimicrobial drug dose, route, timing of administration, and duration [16]. SAP is normally prescribed to most patients before surgery at the Bo government hospital. The nurse anaesthetist administers intravenous ampicillin to the patient just before surgery (within 120 min) and it is regularly supplied as a free healthcare drug for the maternity unit. This is recorded in the anaesthetist’s operation notes. In contrast, herniorrhaphy patients must buy their prescribed drugs as they are not covered by the free healthcare category. The obstetrician, surgeon, or surgical community health officer prescribes postoperative medications for the patient, usually for 24–48 h as postprocedural antibiotic prophylaxis. It may be a shorter period for patients who underwent hernia repair. The intravenous antibiotic regimen of ampicillin, gentamicin, and metronidazole is administered within the first 48 h. Intravenous antibiotics are changed to oral ones such as amoxicillin, augmentin, or ampiclox which are prescribed for five days, with the patients then discharged.

### 2.7. Variables, Data Collection and Validation

The study variables included presence of SSI, type of surgical procedure (CS or herniorrhaphy), socio-demographic and clinical variables, type of antibiotics prescribed, and timing and method of antibiotics (pre, during, or postoperatively; intravenously or orally).

For this study, an SSI was defined as any surgical wound infection as a result of CS or herniorrhaphy occurring during a patient’s admission up until 30 days after surgery. The rate was calculated as the number of SSI cases as a percentage of the total number of surgical procedures during the surveillance period. The diagnosis of SSI was based on clinical assessment by the physician as culture and sensitivity were unavailable. The following clinical features were considered as SSIs: presence of purulent (pus) discharge coming from the wound, unexpected redness or pain, fever, or other signs of sepsis. 

Data from patient registers, the wound dressing book, and individual patient medical records were collected and double entered into EpiData software by the principal investigator supported by three data clerks. Data were collected over four months, double entered, and validated using EpiData (version 3.1, EpiData association, Odense, Denmark).

### 2.8. Analysis

Data analysis was performed using EpiDataStat analysis (v2.2.2.187) software. Descriptive data was summarised using median and interquartile ranges, and also frequencies and proportions.

### 2.9. Ethics

National ethics approval was received from the Sierra Leone Ethics and Scientific Review Board, Freetown, Sierra Leone. International ethics approval (EAG number: 14/21) was given by the Union Ethics Advisory Group of the International Union against Tuberculosis and Lung Disease, Paris, France.

## 3. Results

### 3.1. Characteristics of the Population

A total of 681 patients who underwent surgery at the Bo government hospital were included in the study. During the study period, 599 (88%) had CSs and 82 (12%) had herniorrhaphies. The majority of study patients (88%) were females. The median age among CS patients was 25 years (interquartile range: 20–30 years) and of herniorrhaphy patients, 40 years (interquartile range: 28–55 years). Naturally, all CS patients were female, while 97.6% of the herniorrhaphy patients were male. Most patients, 527 (77%), were married, while 58% lived in urban areas and 42% lived in rural areas. The average hospital stay duration was less than one week.

Most of the surgeries 582 (86%) were emergencies, of which 541 (93%) were emergency CSs. Of the pregnant women who had emergency CS, 243 (45%) presented on admission with obstructed labour. Among the total number of pregnant women, 31% were referred from PHUs. Most patients, 552 (81%), stayed in the hospital for less than seven days.

### 3.2. Surgical Site Infections

The SSI incidence following both CS and herniorrhaphy was 6.7%. Of the 599 patients who underwent CS, 45 (7.5%) developed SSI, while among the 82 herniorrhaphy patients, only 1 (1.2%) developed SSI. The flowchart is shown below in Figure 1.

Among patients with an SSI, 98% were from CSs, 40 (87%) had an emergency CS, and 19 out of 46 (41.3%) stayed in the hospital for less than seven days; the rest stayed for more than one week.

### 3.3. Timing and Type of Antibiotics Administered

Table 1 shows the timing of antibiotics administration. Among CS patients, 15% had preoperative antibiotics only and 70% had both pre and postoperative antibiotics, for a total of 85% who received preoperative antibiotics. About 71% of herniorrhaphy patients also received antibiotics pre and postoperatively.

Table 2 shows the antibiotics used for SAP as well as those prescribed in the postoperative period. Ampicillin and metronidazole were the most commonly used both pre and postoperatively, while amoxicillin was most commonly prescribed orally postoperatively.

## 4. Discussion

This study revealed an SSI rate of 7.5% for CS patients and 1.2% for herniorrhaphy patients at a regional referral hospital in Sierra Leone, adding up-to-date surveillance data for the country. This difference would be expected given the higher risk of infection among emergency CS operations, while most herniorrhaphies are planned, low-risk surgical procedures. This information is important as WHO recommends regular surveillance for SSIs as a way to monitor the results of implementation of IPC procedures. According to Delamou et al. (2019), SSI incidence in 10 maternity facilities in Guinea was 5% in 2015 when substantial IPC measures were implemented during the Ebola outbreak [17]. The SSI incidence in this study at the Bo government hospital maternity unit was higher at 7.5%. However, it was lower than the post CS infection rate of 10.9% in a study conducted at the PCM hospital in Sierra Leone by Di Gennaro et al. (2020) [11].

The incidence of SSIs was lower than for a previous study in Sierra Leone. This is likely because IPC measures were put in place as part of lessons learnt from the Ebola epidemic (2014–2016) and reinforced by the COVID-19 pandemic in 2020. WHO and NIPCU have been supporting IPC practices in hospitals throughout Sierra Leone, and this result reinforces the WHO position that good IPC practices reduce SSIs [6].

The study also showed that 85% of CS and 70% of herniorrhaphy patients received SAP preoperatively, representing a good compliance rate with WHO guidelines. It is also possible that the high SAP compliance rate contributes to the relatively low incidence of SSI. However, 17% of patients did not receive SAP before the procedure. This might relate to the emergency status of these patients and their immediate admission to the theatre, leaving no time for SAP administration.

In a study conducted at a referral hospital in Nepal to assess compliance with their National Antibiotic Treatment Guidelines (NATG), overall compliance was achieved in 75% of patients. About 3% of the patients developed an SSI, of which 35% did not have SAP administered according to the NATG [18].

At the same time, postoperative antibiotics were administered to 85% of CS patients and 100% of herniorrhaphy patients. A systematic review and meta-analysis undertaken by de Jonge et al. (2020) found that there was no benefit in prescribing postoperative antibiotic prophylaxis, as it did not reduce the incidence of SSIs [19]. At Bo government hospital, postoperative antibiotics were still routinely used for both complicated surgeries including emergency CSs and low-risk surgery such as herniorrhaphy. WHO guidelines recommend against the routine use of postoperative antibiotics due to the risk of developing AMR.

To explain the high use of SAP postoperatively, we suggest that some surgeons and obstetricians may have prescribed them because they felt that IPC measures were not entirely adequate and that, clinically, their patients were at risk of developing SSIs. Some surgeons may not have appreciated improvements in IPC that have occurred following Ebola and during the COVID-19 response, or they may have been unaware of the WHO guidelines.

In both surgical procedures, the most commonly used IV prophylactic antibiotics were ampicillin and metronidazole, while oral amoxicillin was most commonly used postoperatively. Gentamicin and ceftriaxone were used almost a quarter of the time. These antibiotics are supplied by the government and have activity against many common susceptible pathogens. However, it is important to note that it is unknown whether these antibiotics were truly appropriate for this context since culture and sensitivity testing was unavailable. This points to the need to develop laboratory services to accommodate this testing. This study had several strengths. First, the cohort included all patients who underwent CS or herniorrhaphy during the study period. Second, the data were collected routinely by staff of the Bo government hospital supervised by the principal investigator (PI), who was the District Medical Officer, as well as the WHO and NIPCU teams. Third, the data clerks who entered data for four months were well trained and supervised by the PI, thus ensuring data quality. Fourth, standard definitions and classifications for eligibility by the WHO SSI surveillance protocol were followed. Fifth, the study followed the Strengthening the Reporting of Observational Studies in Epidemiology (STROBE) guidelines in its presentation [20].

There are limitations in this study. First, SSI surveillance covered only 30 days post surgery. It is possible that some women may have developed SSIs after the 30 days or may have failed to report complications once discharged from hospital or sought care elsewhere. These cases may not have been captured, leading to an underestimation of SSIs. Second, we were unable to collect information for some patients from the female surgical ward as their charts were destroyed in error; however, there were only a few women who had elective herniorrhaphy. Third, the diagnosis of SSI was based on clinical assessment by the physician because specimen microbiology testing was unavailable. Fourth, since only one tertiary referral hospital was studied, the findings on SAP procedures and antibiotic use may not be generalisable for the country. Fifth, we were not able to determine whether postoperative antibiotics were provided as treatment for an SSI or as part of local SAP.

There are a number of operational implications from this study. First, there should be a review of NATG guidelines for Sierra Leone with the addition of the correct method of administering SAP, based on WHO recommendations. Use of postoperative antibiotics should be limited to documented SSIs. Second, education for surgeons, obstetricians, surgical community health officers, and anaesthetists should be a priority since overuse of antibiotics is not only expensive but contributes to AMR, leading to increased morbidity and longer hospitalisation. Education should ideally be part of an overall programme of antibiotic stewardship developed for the hospital. Third, continued improvements in hospital IPC will likely continue to reduce SSIs and build confidence among surgeons that postoperative antibiotics are not needed in most cases. Fourth, laboratory services need to be developed to include culture and sensitivity to be sure the antibiotics chosen are appropriate. Locally developed antibiograms would reduce the use of ineffective antibiotics and AMR. Finally, in keeping with WHO recommendations for ongoing surveillance of SSIs and antibiotic use, this type of study should be repeated regularly to track improvements over time.

## 5. Conclusions

This study found an SSI rate of 7.5% for CS surgery and 1.2% for herniorrhaphy cases in a regional referral hospital in Sierra Leone. Although almost (85%) cases received SAP before surgery, coinciding with WHO recommendations, 70% of both complicated and low-risk cases received postoperative antibiotics. This pattern of overuse can be addressed by locally developed guidelines based on WHO recommendations combined with supportive education for surgeons, obstetricians, anaesthetists, medical doctors, nurses, midwives, and surgical community health officers. The development of laboratory services for culture and sensitivity and further IPC progress should contribute to a reduction in SSIs.

## Figures and Tables

**Figure 1 ijerph-19-04048-f001:**
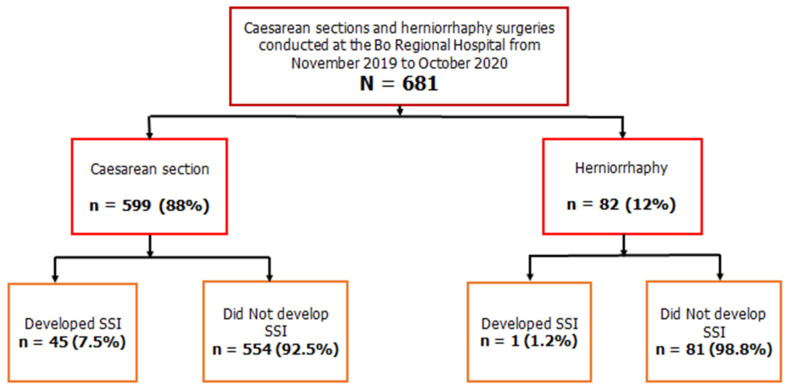
Flowchart showing the number and proportion of surgical site infections (SSI) among surgeries performed by type in the maternity and surgical units, Bo regional hospital, November 2019 to October 2020.

**Table 1 ijerph-19-04048-t001:** Timing of antibiotics administered in patients who underwent Caesarean section and herniorrhaphy surgeries at the Bo regional hospital, Sierra Leone, November 2019 to October 2020.

Antibiotics Administered	Type of Surgical Procedure			
	Caesarean Section		Herniorrhaphy	
	n	(%)	n	(%)
**Timing**				
Preoperative only antibiotics	88	(14.7)	0	(0)
Postoperative only antibiotics	94	(15.7)	24	(29.3)
Both pre and postoperative antibiotics	417	(69.6)	58	(70.7)
**Total**	599	(100)	82	(100)

**Table 2 ijerph-19-04048-t002:** Choice of antibiotics used in patients who underwent Caesarean section and herniorrhaphy surgeries at the Bo regional hospital, Sierra Leone, November 2019 to October 2020.

Antibiotics	Type of Surgical Procedure			
	Caesarean Section		Herniorrhaphy	
	n	(%)	n	(%)
**Choice of Antibiotics**				
Ampicillin	549	(92)	57	(70)
Gentamicin	158	(26)	3	(4)
Metronidazole	389	(65)	75	(92)
Ceftriaxone	121	(20)	32	(39)
Amoxicillin	552	(92)	42	(51)
Other antibiotics	33	(6)	15	(18)

## Data Availability

All data generated or analyzed during this study are available from the corresponding author (R.C.-M.) upon reasonable request. The dataset used in this paper has been deposited at DOI 10.6084/m9.figshare.19203863 and is available under a CC BY 4.0 licence.

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
