# Peer review of "Incidence of Surgical Site Infection and Use of Antibiotics among Patients Who Underwent Caesarean Section and Herniorrhaphy at a Regional Referral Hospital, Sierra Leone"

_ijerph, 2022, doi:10.3390/ijerph19074048_

Round 1

Reviewer 1 Report

Dear Authors,

I have been delighted in reading this interesting article on the incidence of surgical site infection. Authors, reported a rate of SSIs of 7.5% for CS surgery and 1.2% for herniorrhaphy cases in a regional referral hospital in Sierra Leone, that are good results depsite the lack on national guidelines on perioperative SAP and the lack of resources such as laboratory services to for culture and sensitivity.

This Study highlights all the notable results achieved in the field of SSI prevention also compared with other studies conducted in Sierra Leone.

The manuscripts results well-written; however there some issues and concerns.

In MandM authors describe all the geopolitical picture of Sierra Leone that is quite useless for the aim of the study.

In Results there are some typos that should be addressed.

The % of females patients (88%) is the same for both CSs and total patients.

Description of results is quite clear and well-written, but in this study there is no consistent analysis between features groups.

I think that this could be more interesting if a comparison in term of results was conducted between groups of patietnt treated with different Abs or by looking for prognostic factors. 

Moreover, all the limitations declared are too detrimental for the quality of the study. Authors should solve some of the reported limitations such as improving the surveillance.

Author Response

Response to Reviewer 1

Thank you very much for reviewing the manuscript and your insightful comments. We have revised the manuscript in line with your suggestions. We have provided a point-by-point response to your comments and suggestions. Your comments are highlighted in bold and our responses follow using red fonts and bullets.

1st Reviewer comment 1

I have been delighted in reading this interesting article on the incidence of surgical site infection. Authors, reported a rate of SSIs of 7.5% for CS surgery and 1.2% for herniorrhaphy cases in a regional referral hospital in Sierra Leone, that are good results depsite the lack on national guidelines on perioperative SAP and the lack of resources such as laboratory services to for culture and sensitivity.

This Study highlights all the notable results achieved in the field of SSI prevention also compared with other studies conducted in Sierra Leone.

The manuscripts results well-written; however there some issues and concerns.

In MandM authors describe all the geopolitical picture of Sierra Leone that is quite useless for the aim of the study.

Response

Thank you very much for critiquing and appreciating this work. Description of the geopolitical picture of the country and district is part of the standard writing format in the SORT – IT programme. However, based on your advice, the geopolitical description of Sierra Leone has been reduced.

1st Reviewer comment 2

In Results there are some typos that should be addressed.

Response

The typographical error in the result section have been corrected.

1st Reviewer comment 3

The % of females patients (88%) is the same for both CSs and total patients.

Response

There were 599 cases of Caesarean sections (88%) out of a total of 681surgical cases. Naturally, all of them were females. The herniorrhaphy cases in the male ward were included. We could not capture the data from the few female hernia cases since their clinical charts in the female ward were destroyed, hence the 88% coincidence.

1st Reviewer comment 4

Description of results is quite clear and well-written, but in this study there is no consistent analysis between features groups.I think that this could be more interesting if a comparison in term of results was conducted between groups of patietnt treated with different Abs or by looking for prognostic factors. 

Response

In section 3.3 table 2, both groups of patients who had CS or a herniorrhaphy received the same type of antibiotics since these were readily prescribed and available. Ampicillin, metronidazole and amoxicillin were more prevalent in both groups.

1st Reviewer comment 5

Moreover, all the limitations declared are too detrimental for the quality of the study. Authors should solve some of the reported limitations such as improving the surveillance.

Submission Date          20 February 2022

Date of this review        07 Mar 2022 20:15:22

Response

Your point on the limitations was noted however patients have various methods of illness seeking behaviors. It is still possible that some patients had late complications and went to another health facility or traditional healers. These ones will be missed since it was not reported.

Thank you

Reviewer 2 Report

The study is only descriptive in regard of antibiotic prophylaxis in a particular hospital of Sierra Leona, which is not of great interest to the general medical community, other than the locals. Hence, it does not add any particular knowledge of interest.

The study methods have several defects, such as the use of postoperative antibiotics in most part of the cases, being both pre and postoperative type of antibiotic, and use timing a decision entirely up to the surgeon´s choice. Also, the 1st choice of recommended preop antibiotics are 1st generation cephalosporins, and ampicillin was used in most part of this case series.

Pre-operative antibiotics were not used in many CS cases due to the urgent need of surgery, and it is not clear is abdominal prep of the patients was identical to the ones with a scheduled surgery. In this regard, it would have been interesting to compare, for example, the rate of post-op wound infection in urgent vs. scheduled cases for both types of surgical procedures (CS and hernioplasty).

As an attempt to homogenize the data, I would discard the patients with post-op antibiotic usage.

As the authors state, there were several patients with lack of follow-up due to varied reasons.

Author Response

Response to Reviewer 2

Thank you very much for reviewing the manuscript and your insightful comments. We have revised the manuscript in line with your suggestions. We have provided a point-by-point response to your comments and suggestions. Your comments are highlighted in bold and our responses follow using red fonts and bullets.

2nd Reviewer comment 1

The study is only descriptive in regard of antibiotic prophylaxis in a particular hospital of Sierra Leona, which is not of great interest to the general medical community, other than the locals. Hence, it does not add any particular knowledge of interest.

Response

Thank you very much for critiquing this work. This work as you mentioned is largely descriptive and the information detailed in this article is of interest to the policy makers, healthcare providers and the public. In this Anti-Microbial Resistance (AMR) study, apart from SSI surveillance, we wanted to know the level of compliance with the WHO guidelines on Surgical Antibiotic Prophylaxis or preoperative antibiotics, the level of overuse of antibiotics especially postoperative antibiotics and the need for a National Antibiotic Treatment Guideline adapted from the WHO SAP guideline. The method observed in this referral hospital is widely practiced in other hospitals around the country.

2nd Reviewer comment 2

The study methods have several defects, such as the use of postoperative antibiotics in most part of the cases, being both pre and postoperative type of antibiotic, and use timing a decision entirely up to the surgeon´s choice. Also, the 1st choice of recommended preop antibiotics are 1st generation cephalosporins, and ampicillin was used in most part of this case series.

Response

These points are noted. The current antibiotic prescription pattern was observed and the data collected and analysed as part of SSI surveillance. From these findings, we will be able to recommend cost-effective and sustainable solutions that will lead to considerable reduction in AMR. The broad spectrum IV ampicillin is widely available and cheaper than IV Cephazolin. Also, it is supplied to the government hospitals as part of free health drugs so it is given free to the pregnant women.

2nd Reviewer comment 3

Pre-operative antibiotics were not used in many CS cases due to the urgent need of surgery, and it is not clear is abdominal prep of the patients was identical to the ones with a scheduled surgery. In this regard, it would have been interesting to compare, for example, the rate of post-op wound infection in urgent vs. scheduled cases for both types of surgical procedures (CS and hernioplasty).

Response

Combining the two categories (pre-operative only and both pre- and post-operative antibiotics), preoperative antibiotics were given to 84.3% of CS patients and 70.7% of herniorrhaphy patients. This is a relatively good compliance rate. This is already in the 3rd paragraph of section 4. Standard abdominal preparations were done for both emergency and elective surgeries.

From the data on just the SSI cases which were 46 in number, 97.8% of SSIs were from CSs and one (2.2%) was from herniorrhaphy. Elective cases that developed SSI was 13% and the rest (87%) were from emergency cases. Part of this was in the manuscript and it has been further edited based on your advice in section 3.2 second paragraph.

2nd Reviewer comment 4

As an attempt to homogenize the data, I would discard the patients with post-op antibiotic usage. As the authors state, there were several patients with lack of follow-up due to varied reasons.

Submission Date                      20 February 2022 

Date of this review                    03 Mar 2022 16:55:03

Response

We kindly prefer the data on post-operative antibiotics be kept so that healthcare providers can understand the level of overuse of antibiotics in this regional referral hospital and thus take our recommendations seriously.

Round 2

Reviewer 1 Report

I really appreciate the effort made by authors in aswering to all my concerns. Unfortunately not all of those have been properly solved but I believe that the text deserve a chance for publication after all. 

Reviewer 2 Report

Although I emphasize with you, being from a country with serious public health problems myself, I think that if you want to expose a public health problem, so health policies can be changed, a medical journal is not the place where you may do this. Unfortunately, I don’t see how can be a local health problem of interest for the rest of the health community. Kind regards.